# CoC-VLA: Delving into Adversarial Domain Transfer for Explainable Autonomous Driving via Chain-of-Causality Visual-Language-Action Model

**Dapeng Zhang**[1,2]**, Fei Shen**[2]***, Rui Zhao**[1]**, Yinda Chen**[3]**, Peng Zhi**[1]**, Chenyang Li**[1]**,
**Rui Zhou**[1]**, Qingguo Zhou**[1]***

[1] Lanzhou University, China
[2] National University of Singapore, Singapore
[3] University of Science and Technology of China, China
[*] Corresponding authors
{zhangdp22, zhaorui, zhip21, lchenyang2024, zr, zhouqg}@lzu.edu.cn,
shenfei29@nus.edu.sg, cyd0806@mail.ustc.edu.cn

## Abstract

Autonomous driving represents a prominent application of artificial intelligence. Recent approaches have shifted from focusing solely on common scenarios to addressing complex, long-tail situations such as subtle human behaviors, traffic accidents, and non-compliant driving patterns. Given the demonstrated capabilities of large language models (LLMs) in understanding visual and natural language inputs and following instructions, recent methods have integrated LLMs into autonomous driving systems to enhance reasoning, interpretability, and performance across diverse scenarios. However, existing methods typically rely either on real-world data, which is suitable for industrial deployment, or on simulation data tailored to rare or hard case scenarios. Few approaches effectively integrate the complementary advantages of both data sources. To address this limitation, we propose a novel VLM-guided, end-to-end adversarial transfer framework for autonomous driving that transfers long-tail handling capabilities from simulation to real-world deployment, named CoC-VLA. The framework comprises a teacher VLM model, a student VLM model, and a discriminator. Both the teacher and student VLM models utilize a shared base architecture, termed the Chain-of-Causality Visual–Language Model (CoC VLM), which integrates temporal information via an end-to-end text adapter. This architecture supports chain-of-thought reasoning to infer complex driving logic. The teacher and student VLM models are pre-trained separately on simulated and real-world datasets. The discriminator is trained adversarially to facilitate the transfer of long-tail handling capabilities from simulated to real-world environments by the student VLM model, using a novel backpropagation strategy. Experimental results show that our method effectively bridges the gap between simulation and real-world autonomous driving, indicating a promising direction for future research.

## 1 Introduction

Autonomous driving has advanced significantly over the past decades, attracting interest from both commercial and academic sectors. It has evolved from simple trajectory tracking into a complex, integrated system. Typically, an autonomous driving system consists of several modules, including environmental perception and decision-making. These systems may rely on rule-based methods for navigating familiar roads or imitate human behavior to manage a wide range of driving scenarios. Prevailing autonomous driving approaches primarily focus on achieving breakthroughs on challenging benchmarks and are typically trained and evaluated on datasets collected in real-world

---

[*]Corresponding authors

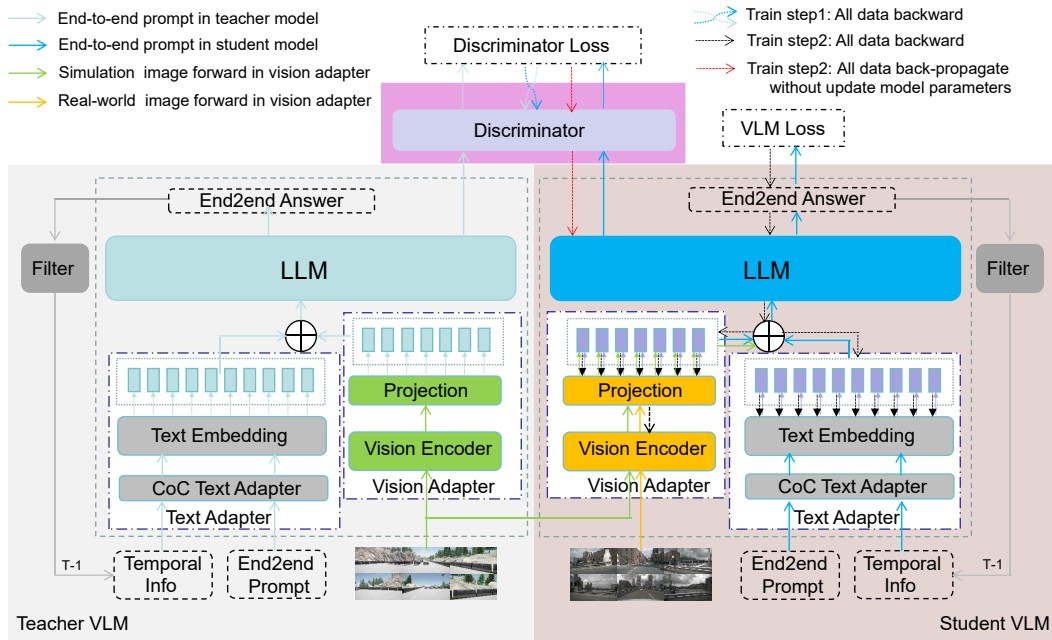

Figure 1: Overview of the proposed framework. The framework comprises a teacher VLM model (based on the simulation CoC VLM baseline), a student VLM model (based on the real-world CoC VLM baseline), and a discriminator. The teacher and student VLM models share the same architecture but are trained separately on simulation and real-world datasets, respectively. The discriminator is employed to facilitate the adversarial transfer of capabilities from the simulation domain to the real-world domain. Train step1 refers to the adversarial training step1. Train step2 refers to the adversarial training step2.

environments. However, such systems may fail in previously unseen cases, such as rare accidents or unexpected human behaviors, prompting the development of simulated dataset benchmarks to evaluate performance under rare or hard case scenarios.

To ensure robust performance across diverse environments, researchers have adopted data-driven end-to-end autonomous driving methods. These methods enhance system integrity by eliminating accumulated errors. Moreover, the widespread adoption of end-to-end models enables faster inference and lower resource consumption [1, 2]. By processing input from surrounding images and/or LiDAR data, end-to-end systems generate final trajectories and/or control signals. However, these systems operate as black boxes, offering no interpretability or explanation for their decisions, thereby raising ethical and legal concerns. Furthermore, they lack mechanisms for human interaction, limiting their applicability in advanced autonomous driving.

To address the black-box nature of end-to-end autonomous driving, numerous studies have explored the integration of large language models (LLMs) to enhance vehicle interpretability, controllability, and robustness. LLMs have demonstrated remarkable capabilities when trained on large-scale datasets, which are highly desirable in autonomous driving systems. Additionally, LLMs exhibit strong generalization capabilities, enabling them to handle unseen scenarios and unfamiliar environments. Among LLM-based autonomous driving approaches, several are trained and evaluated on real-world datasets for industrial deployment. Inspired by [3, 4], DriveGPT4 [5] represents a pioneering effort to leverage LLMs for interpretable end-to-end autonomous driving. It accepts multi-frame sequences and instruction prompts, and outputs vehicle control signals. Experimental results on real-world datasets demonstrate the effectiveness of this method. LLMs offer a capability absent in traditional autonomous driving systems: they can interpret textual descriptions and translate driving instructions into actionable control signals. Moreover, LLM-based approaches aim to address long-tail challenges in autonomous driving by employing diverse modeling strategies and utilizing various sensor modalities [6–8]. However, LLMs trained on real-world datasets often fail to capture critical uncommon cases such as traffic accidents, non-compliant driving behaviors, and pedestrian intrusions. To address this limitation, some approaches are trained on simulated data and evaluated in virtual environments. For example, [9] trains a language-based autonomous driving model using

data collected in the CARLA simulator [10], and performs closed-loop evaluations within simulated environments. However, these methods are tested on simulators, where the generated control signals cannot be executed in real-world conditions. In summary, the gap between real-world and simulated datasets remains an underexplored area of research.

Furthermore, to incorporate temporal information from historical frames, existing methods typically feed image sequences directly into the LLM [9]. This significantly increases token length and computational resource consumption. Therefore, an efficient and lightweight temporal aggregation strategy is essential.

In this paper, we propose a novel VLM-based model for explainable end-to-end autonomous driving that not only effectively transfers capabilities learned from simulated hard cases to real-world applications, but also integrates temporal information and end-to-end outputs via a chain-of-causality policy. The model takes surrounding image pairs and driving instructions in natural language, and predicts CoC answers. The overall architecture is illustrated in Fig. 1. Our method comprises two base models: a teacher VLM model and a student VLM model, as well as a discriminator. The teacher VLM model is trained on synthetic data to acquire the ability to handle rare and challenging scenarios, and transfers this knowledge to the student VLM model via adversarial learning with the discriminator. The training process is non-trivial, requiring both a pre-trained VLM model and multiple stages of training. During inference, only the student VLM model is deployed in real-world scenarios. Furthermore, we reproduce several existing methods and compare them with ours using a public benchmark dataset. Experimental results demonstrate that our method achieves an excellent result.

The contributions of this work are summarized as follows:

- We present the first VLM-based autonomous driving model capable of transferring the ability to handle uncommon scenarios from simulation to real-world, thereby bridging the gap between simulated and real-world environments.

- We introduce a novel discriminator that learns the domain gap between simulation and real-world data, enabling effective knowledge transfer from the teacher to the student VLM model.

- We propose a back-propagation strategy that enhances the convergence stability of the adversarial training process.

- We develop a chain-of-causality policy that connects temporal information to CoC answers, enabling chain-of-thought reasoning to model deep driving logic.

- We conduct extensive experiments on the nuScenes-VLM dataset, demonstrating that our approach significantly outperforms existing methods.

## 2  Related Works

### 2.1  End-to-End Autonomous Driving.

Traditional autonomous driving methods suffer from complex module designs and limited interaction between modules, often resulting in error accumulation. To address these challenges, researchers have proposed end-to-end autonomous driving methods that unify previously separate modules into a cohesive framework. Typically, these methods integrate perception, mapping, prediction, and planning sub-tasks into a single model. For instance, UniAD [1] integrates features from the perception and prediction modules and generates ego-vehicle planning trajectories using a transformer architecture. Building on BEVFormer [11], VAD [2] extracts BEV features and regresses planning trajectories through multiple interactions and constraints. FusionAD [12] also predicts ego trajectories on BEV maps constructed from both camera data and LiDAR point clouds. VAD2 [13] introduces a vectorized encoding approach to model the probabilistic distribution of trajectories, demonstrating superior closed-loop performance. Additionally, Ego-MLP and BEV-Planner [14, 15] extensively explore the impact of ego-vehicle status to enhance trajectory planning accuracy. ThinkTwice [16] retrieves encoder features near the predicted coordinates and refines coarse-grained positions and actions. Inspired by these advancements, ReasonNet [17] utilizes global and temporal representations of driving scenarios to enhance feature extraction. Notably, TCP [18] integrates trajectory and control action predictions into dedicated branches to improve overall driving robustness. Roach [19] employs reinforcement learning to distill a final agent capable of interacting effectively with dynamic environments. In contrast to prior methods, SparseDrive [20] employs sparse feature sampling alongside a hierarchical planning strategy to generate rational and efficient planning outputs. Since open-loop evaluations do not account for dynamic responses from surrounding vehicles, researchers

have introduced closed-loop evaluation metrics, such as Driving Score, Route Completion, and Infraction Score, to more accurately assess and optimize their models [21, 10].

## 2.2 LLM for Autonomous Driving

In recent months, the emergence of large language models (LLMs) [22, 3, 4] has led researchers to extend them into vision-language models (VLMs) [23, 24], which integrate textual and visual data for richer content representation. In the field of autonomous driving, researchers have begun using LLMs and VLMs to enhance overall system performance. For example, DriveGPT4 [5] uses multi-modal input data to generate expected control signals for the vehicle. Another method integrates LLMs into autonomous driving frameworks to generate action recommendations along with detailed explanations [9]. However, the control actions predicted by these methods often fall short of real-world navigation demands. Consequently, researchers are increasingly focusing on generating textual descriptions of driving actions to offer interpretable and contextually appropriate explanations. For instance, ContextVLM [6] incorporates diverse environmental contexts to enhance robustness across a range of scenarios. Instead of relying on camera data, LiDAR-LLM [7] utilizes raw LiDAR inputs and employs a three-stage training strategy to align 3D modalities with the LLM embedding space, thereby enhancing spatial understanding for autonomous driving. DriveMM [8] processes diverse inputs, such as images and multi-view videos, to pre-train a baseline model, which is then refined to improve generalization in vehicle control. To improve temporal representation, LaVidaDrive [25] introduces a Query-aware Token Selection module, a Spatial-Temporal Token Recovery and Enhancement module to optimize both efficiency and performance. To address VLM limitations in spatial reasoning, DriveVLM [26] integrates specialized reasoning modules for scene understanding and hierarchical planning. Additionally, DriveMLM [27] introduces a behavior planning module to generate optimal driving decisions with interpretable justifications. Notably, recent approaches incorporate reinforcement learning to enhance multi-modal planning capabilities [28].

## 2.3 Domain Transfer

Domain transfer learning aims to build models capable of performing tasks in a target domain by leveraging knowledge learned from a source domain. The method proposed in [29], implemented using a deep learning strategy, achieves effective domain adaptation across several classification datasets. To explore domain-specific characteristics, [30] explicitly extracts image representations partitioned into two subspaces: one private to each domain and the other shared across domains. Given the importance of pre-training in transfer learning, some methods leverage it to enhance adversarial robustness compared to other approaches [31]. Most existing methods align the fully connected layers in neural networks, while convolutional layers, which typically encode critical low-level domain knowledge, are often left unmodified. This limitation restricts the effectiveness of domain discrepancy reduction. To address this, [32] proposes an attention alignment mechanism on convolutional layers to better minimize discrepancies between domains. MetaAlign [33] introduces a novel meta-optimization strategy that maximizes gradient-based learning during training. Additionally, some researchers have proposed asymmetric training schemes that align target domain features more closely with those from the source domain [34]. [35] leverages object attributes to facilitate robotic grasping and rapid adaptation across domains. Inspired by [36], which simultaneously trains a generative model to learn data distribution and a discriminative model to estimate sample likelihoods, Pix2Pix [37] extends the GAN-based strategy to image-to-image translation tasks, demonstrating the effectiveness of the discriminator in image generation. Subsequently, numerous methods have emerged to address diverse application scenarios, including PatchGAN, DTSGAN, and RFGAN [38–40].

## 3 Methods

As illustrated in Fig. 1, the proposed architecture consists of three components: a Teacher Visual Language Model (Teacher VLM), a Student Visual Language Model (Student VLM), and a Visual Language Model Discriminator. Both the Teacher and Student VLMs share a common base architecture, referred to as the Chain-of-Causality Visual Language Model (CoC VLM), which processes multi-view image pairs, end-to-end prompts, and historical instructions from previous frames to generate end-to-end outputs. The Teacher VLM is trained on simulated data to address diverse and rare scenarios, such as pedestrian trespassing, driving violations, and traffic accidents. In contrast, the Student VLM is trained on real-world data and serves as the final inference model, enabling the transfer of knowledge from simulated to real-world contexts.

## 3.1 Chain-of-Causality Visual Language Model

Since the Teacher VLM is primarily designed to transfer the capability of handling hard and challenging cases to the Student VLM, both VLMs are constructed using the same base architecture but with different parameter sets. We design the Chain-of-Causality Visual Language Model as the shared backbone for both the Teacher and Student VLMs. Based on extensive experimentation, LLaVA-v1.5 [41] was selected as the pre-trained VLM. The CoC VLM is primarily based on LLaVA [41] and comprises four modules: Text Adapter, Vision Adapter, LLM Brain, and CoC Answer. Compared to the original LLaVA, the CoC VLM introduces several enhancements: (1) A novel Chain-of-Causality Text Adapter aggregates simplified answers from the previous frame and current instruction prompts, thereby incorporating historical context and enhancing temporal causal reasoning. (2) A filter is employed to simplify the LLM-generated responses, effectively reducing token length. (3) A dedicated CoC answer generation module is introduced to streamline output formatting.

### 3.1.1 Text Adapter

The Text Adapter comprises two components: the Chain-of-Causality (CoC) Text Adapter and the Text Embedding module. The CoC Text Adapter aggregates temporal information from the previous frame along with the current end-to-end prompt. Unlike existing methods such as LMDrive [9], which utilize all historical sensor data to encode temporal information, our method selectively incorporates simplified LLM outputs from the previous frame to enhance temporal consistency. This approach significantly reduces token length and computational resource consumption. The proposed Text Adapter is both simple and efficient. An illustrative example is provided below:

- **Temporal Instruction:** In the previous frame, a white car ahead is moving away from the ego vehicle, and the vehicle continues in its current driving state.

- **End-to-End Prompt:** What are the objects around the ego vehicle? What is the moving status of this object? What is the next action of the closest object? Will thread ego vehicle's safety? What are safe actions to take for the ego vehicle? Predict the future motion of the ego vehicle.

Given the Temporal Instruction and End-to-End Prompt, we tokenize and concatenate the texts into textual tokens, which are then embedded using the same embedding module as LLaVA [41].

### 3.1.2 Vision Adapter

The Vision Adapter transforms image data into tokens, as illustrated in Fig. 1. To emulate human driving behavior, which relies solely on a 2D visual perspective, we directly stack six surrounding camera images as input. No extrinsic or intrinsic parameters are utilized, nor are the images transformed into BEV features, as human drivers also do not rely on such representations. This approach contrasts with several existing end-to-end autonomous driving methods. Following the guidance of LLaVA-V1.5 [41], we adopt the pre-trained CLIP image vision tower as our vision encoder and apply a projection module to convert image features into tokens, which are subsequently integrated into the textual prompt.

### 3.1.3 LLM Brain

The LLM Brain employs a specialized chain-of-thought reasoning mechanism to infer deep driving logic. This module processes image tokens generated by the vision adapter and combines them with text instructions from the text adapter to comprehend driving scenarios and generate tokens for the next step. Similar to LLaVA-v1.5 [41], we adopt the LLaMA [4] language model as our LLM Brain. Leveraging its pre-trained weights and fine-tuned components, our model achieves convergence as we expected. The LLM Brain outputs end-to-end autonomous driving answers to manipulate the ego vehicle.

### 3.1.4 CoC Answer

After processing by the LLM Brain, the output token sequences are decoded using a tokenizer decoder to generate CoC answers. Inspired by [42], our CoC answer integrates perception, prediction, and planning into a causal structure, which we represent as a causal chain. Specifically, perception determines the future actions of surrounding objects, which subsequently influence the ego vehicle's motion. Causal reasoning is applied throughout the entire CoC answer. Furthermore, we design a filter to summarize and cache the CoC answer, converting it into a concise instruction for the subsequent frame. We provide an example below. This filter removes the reasoning component of the CoC answer and extracts only the final action instruction for the ego vehicle—for example, and reduces the input token size while preserving the most critical information from the previous frame.

- **CoC Answer:** There is a white car in front of the ego vehicle, with coordinates <CAM_FRONT, 1009, 486, 1074, 527>. The white car <CAM_FRONT, 1009, 486, 1074, 527> is accelerating and moving away. There is a traffic light $\cdots$, and there are two pedestrians $\cdots$. In front of the ego vehicle, there is no safety threat in front of the ego vehicle, the ego vehicle should continue moving at the same speed. The future trajectory is (4.7, -0.7), (7.4, -1.3) $\cdots$.

- **Cached Temporal Information:** There is no safety threat, the vehicle maintains its current speed.

### 3.2 Discriminator

#### 3.2.1 Structure Design

This module aims to address the distributional gap between real-world and simulation domains by transferring the performance of the teacher VLM model to align with the expectations of the student VLM model. The discriminator is implemented using transformer architectures. It processes features from both the teacher and student VLM models. Its objective is to minimize the domain gap and adversarially align feature representations for the student VLM model.

#### 3.2.2 Discrepancy Analysis

Let the data spaces of the simulation and real-world domains (referred to as the teacher and student domains, respectively) be denoted as $X^S$ and $X^R$. We denote the distributions that collected data samples from these two domains as $\{x_i^s, p^s(y_i^s|x_i^s)\} \in X^S$ and $\{x_i^r, p^r(y_i^r|x_i^r)\} \in X^R$, $x_i^s$ and $x_i^r$ are the data samples of simulation and real-world domains, $p^s(y_i^s|x_i^s)$ and $p^r(y_i^r|x_i^r)$ represent the conditional label distributions in the source and target domains, respectively. $i$ is the sample index. The discriminator is designed to learn representations that capture the domain shift between the two domains. Since both the teacher and student VLM models project data into feature spaces, the corresponding transformations are:

$$\begin{cases} Z^S = g_s(x_i^s) & x_i^s \in X^S \\ Z^R = g_r(x_i^r) & x_i^r \in X^R \end{cases} \tag{1}$$

where, $Z^S$ and $Z^R$ are the feature space representation of two inputs. $g_s$ and $g_r$ are the corresponding functions of two base models, tasked with preserving rich information relevant to autonomous driving. The discriminator extracts representations in a hypothesized domain space:

$$\begin{cases} H^S = h_s(Z^S) = h_s(g_s(x_i^s)) & x_i^s \in X^S \\ H^R = h_r(Z^R) = h_r(g_r(x_i^r)) & x_i^r \in X^R \end{cases} \tag{2}$$

$H^S$ and $H^R$ denote the output distribution spaces induced by the compositions $h_s \cdot g_s$ and $h_r \cdot g_r$, $h_s$ and $h_r$ are hypothesis functions to unify the feature spaces.

Given the definable difference between domains, we introduce a transformation matrix $T_{s2r}$ to minimize the distributional distance between the simulation and real-world domains. Accordingly, the transformation function between the two domains is defined as:

$$H^R := T_{s2r} \cdot H^S := T_{s2r} \cdot (h_s \cdot g_s) \tag{3}$$

With the relationship between the two domains established, the hypothesis function for the real-world domain is defined as:

$$\hat{y}_i^r = T_{s2r} \cdot h_s(g_s(x_i^r))|h_r : X^R \tag{4}$$

here, the real-world output $\hat{y}_i^r$ is defined by $h_s$, $g_s$ and transfer matrix $T_{s2r}$.

This method aims to quantify the discrepancy between the two domains. Given the two equations defined above, the discrepancy $D_h^\delta$ between two domains $X^S$ and $X^R$ can be expressed by space samples of the two domains. An invariant representation is expected to satisfy $D_h^\delta(X^S||X^R) = 0$. Hence our algorithms learning representations and minimizing the domain error when label distributions differ between source and target domains.

$$\begin{aligned} D_h^\delta(X^S||X^R) := & \underset{x^s \in X^S}{\mathbb{E}} \left( l_s(h_s(g_s(x^s)), y^s) \right. \\ & - \underset{x^r \in X^R}{\mathbb{E}} \left( l_r(h_r(g_r(x^r)), y^r) \right. \end{aligned} \tag{5}$$

These results demonstrate that the hypothesis function $h \cdot g$ captures domain data distributions and directly affects the measured discrepancy. As indicated by these equations, an upper bound is required to characterize the simulation domain, which can be defined as:

$$sup_{x^s \in X^S} \mathbb{E} [l_s(h_s(g_s(x^s)), y^s)] := \rho < \infty \tag{6}$$

Here, $l_s$ is the bounded distance loss, the expected parameter is represented with $\rho$. Given the transformation matrix $T_{s2r}$ linking the two domains, we further explore the formula for discrepancy $D^\delta$:

$$D_h^\delta(X^S||X^R) <= sup(\mathbb{E}_{x^s \in X^S} (h_s(g_s(x^s)) - y^s)$$
$$- \mathbb{E}_{x^r \in X^R} (\hat{T}_{s2r}(h_s(g_s(x^r)) - y^r))) \tag{7}$$

where $sup$ is a supremum, $\hat{T}_{s2r}$ denotes the Fenchel conjugate of a lower semi-continuous convex function. This discrepancy $D_h^\delta$ is a variational formulation of the f-divergence for the convex function $\delta$, thus, $D_h^\delta(X^S||X^R)$ serves as a lower bound estimation of the f-divergence function.

In our method, the hypothesis can clearly explain the foundation of our hypothesis, since the final function is a continuous convex function, which can be optimized with appropriate solvers.

### 3.2.3 Adversarial Optimization

In our VLM-based autonomous driving model, two components must be optimized to effectively transfer knowledge from the simulation domain to the real-world domain. These components are the VLM autonomous driving regression and the domain discrepancy discriminator, denoted as $D_{dis}^\delta(X^S||X^R)$. The discriminator estimates and minimizes the feature distribution discrepancy between two domains $X^S$ and $X^R$. The total optimization problem is formulated as:

$$\min_\delta (\int_0^n l_{VLM}(h_{VLM}(x), y)p(y|x)p(x)dydx + D_{dis}^\delta(X^S||X^R)) \tag{8}$$

where, $l_{VLM}$ is the VLM autonomous driving loss, $h_{VLM}$ is the VLM autonomous driving optimization function. Furthermore, we define an upper bound for this discriminator:

$$d_{dis}^\delta = \int_0^n l_{dis}(h_{dis}(g_{dis}(x^{s,r})), y^{s,r})p(y^{s,r}|x^{s,r})p(x^{s,r})dy^{s,r}dx^{s,r} \tag{9}$$

where $h_{dis} \cdot g_{dis}$ represents the discriminator networks responsible for extracting features from the input data. From the two equations above, we derive the following inequality:

$$D_{dis}^\delta(X^S||X^R) <= \max_\delta d_{dis}^\delta \tag{10}$$

The final optimized function is given by:

$$\min_\delta \int_0^n l_{VLM}(h_{VLM}(x), y)p(y|x)p(x)dydx$$
$$+ \min_\delta \max_\delta \int_0^n l_{dis}(h_{dis}(g_{dis}(x^{s,r})), y^{s,r})p(y^{s,r}|x^{s,r})p(x^{s,r})dy^{s,r}dx^{s,r} \tag{11}$$

As shown in Eq. (11), our model first minimizes the VLM loss functions using finite samples, the second components correspond to discriminator losses, which are optimized using an adversarial strategy with a min-max formulation.

## 4 Experiments

### 4.1 Model Training

The training procedure comprises two stages: pre-training and adversarial training.

### 4.1.1 Pre-training

During pre-training, the teacher VLM model and student VLM model are trained separately using simulation and real-world datasets, respectively. To expedite training, checkpoints from LLaVA-v1.5 [41] are loaded as initialization. Both models accept image frames, temporal instructions, and end-to-end prompts as inputs to fine-tune their respective Chain-of-Causality (CoC) VLMs, thereby effectively aligning instruction, visual, and temporal information. Efficient training is achieved by sampling frames at fixed intervals and applying temporal augmentation through random temporal shifts.

### 4.1.2 Adversarial Training

As illustrated in Fig. 1, the adversarial training process involves multiple steps. Initially, the pre-trained teacher and student VLM models are loaded, and then the following steps are executed. Step 1: The teacher VLM processes simulation data, while the student VLM processes real-world data. Features extracted from both CoC VLMs are subsequently fed into the discriminator. After calculating the discriminator loss, backpropagation is performed to only update exclusively the discriminator parameters.
Step 2: The student VLM model is forward propagated using the real-world dataset, with concurrent involvement of the discriminator. This step optimizes the student VLM model using both its autonomous driving loss and the discriminator loss. During backpropagation, the discriminator propagates gradients without updating its parameters. The student VLM model is updated through backpropagation based on the combined loss.

### 4.2 Main Comparison

We conducted a VQA-related experiment to compare our method with existing LLM-based autonomous driving approaches. As illustrated in Table 1, our method demonstrates significant advantages in VQA performance on the nuScenes-VLM dataset. The LLM-based autonomous driving task is similar to traditional LLM tasks. Our method significantly outperforms the well-known DriveLM [42] in terms of BLEU scores, achieving BLEU-1, BLEU-2, BLEU-3, and BLEU-4 scores of 74.06, 69.33, 63.77, and 58.84, respectively, using the LLaVA-7b backbone. We also trained our student VLM model on a mixed dataset comprising both simulation and real-world data. Besides, we conduct another experiment, we using a two-stage fine-tuning process: first on the simulation dataset, followed by fine-tuning on the real-world dataset. Both approaches yielded inferior results compared to our method. The ROUGE-L performance aligns with our expectations, achieving improvements of 3.38 and 1.01 over DriveLM and our proposed-FinetuneTwice model, respectively. Notably, our method achieves nearly a 5-fold improvement over the competing method in terms of CIDEr score. Specifically, our LLaVA-7b-based method shows a 1.05× improvement in accuracy compared to DriveLM [42]. Furthermore, it yields a Match score of 45.30 and a SPICE score of 51.88, both of which significantly surpass competing methods.

In our autonomous driving quantitative experiments, we aim to evaluate our method's ability to perform open-loop driving on the nuScenes-VLM dataset, focusing on transfer learning from simulation to real-world scenarios and addressing the scarcity of challenging cases in real-world data. As shown in Table 2, our approach demonstrates superior performance on the nuScenes-VLM dataset. Specifically, our method with the LLaVA-7b backbone outperforms the competing method [42] by more than 15.8% and 15.5% in ADE and collision rate, respectively. Notably, our final proposed model achieves improvements of 0.16 and 0.15 over the proposed-FinetuneTwice model. These findings strongly validate the effectiveness of our proposed model architecture.

Table 1: The general performance on our nuScenes-VLM dataset is evaluated using various language metrics. * Indicates reproduced results. Proposed-Mix refers to the approach in which the student VLM model is mix-trained using both simulated and real-world data. Proposed-FinetuneTwice denotes the strategy where the student VLM model is first fine-tuned on simulated data and subsequently fine-tuned on real-world data to obtain the final results.

| Methods | BLEU ↑ | | | | ROUGE-L ↑ | CIDEr ↑ | GPT ↑ | Accuracy ↑ | Match ↑ | SPICE ↑ |
| --- | --- | --- | --- | --- | --- | --- | --- | --- | --- | --- |
| | 1 | 2 | 3 | 4 | | | | | | |
| DriveLM [42]* | 72.64 | 66.77 | 61.02 | 55.13 | 68.32 | 3.62 | 58.21 | 36.67 | 34.59 | 48.95 |
| Proposed-Mix(LLaVA-7b) | 71.98 | 67.31 | 61.57 | 56.11 | 68.60 | 21.07 | 60.22 | 73.13 | 40.38 | 49.98 |
| Proposed-FinetuneTwice(LLaVA-7b) | 72.18 | 68.07 | 61.77 | 58.36 | 70.69 | 21.21 | 61.72 | 73.74 | 41.51 | 49.87 |
| Proposed(LLaVA-7b) | 74.06 | 69.33 | 63.77 | 58.84 | 71.70 | 23.21 | 62.64 | 75.31 | 45.30 | 51.88 |

### 4.3 Ablation Study

### 4.3.1 Key Components Effectiveness

We conduct several ablation studies to evaluate the effectiveness of key component designs, with results presented in Table 3. Five experiments are conducted using different combinations of model components. As shown in Index-1 of Table 3, the teacher VLM model is trained solely on the simulated dataset and evaluated on the nuScenes-VLM dataset. This configuration results in poor

Table 2: The open-loop evaluation of planning performance is conducted on our nuScenes-VLM dataset. This evaluation is based on Average Displacement Error (ADE) and Collision Rate metrics. * Indicates results reproduced on our nuScenes-VLM dataset. Proposed-Mix refers to the approach in which the student VLM model is mix-trained using both simulated and real-world data. Proposed-FinetuneTwice denotes the strategy where the student VLM model is first fine-tuned on simulated data and subsequently fine-tuned on real-world data to obtain the final results.

| Methods | ADE ↓ | Collision Rate (%) ↓ |
|---|---|---|
| DriveLM*[42] | 1.71 | 1.87 |
| Proposed-Mix (LLaVA-7b) | 1.84 | 1.96 |
| Proposed-FinetuneTwice (LLaVA-7b) | 1.60 | 1.73 |
| Proposed (LLaVA-7b) | 1.44 | 1.58 |

performance. Similarly, as shown in Index-2, training the student VLM model exclusively on the nuScenes-VLM dataset yields suboptimal results, achieving an ADE of 1.66 and a collision rate of 1.80%. In Index-3, the student VLM model is trained on a mixture of simulated and real-world datasets. A slight performance degradation is observed compared to Index-2, which can be attributed to the model's attempt to generalize across two distinct domains, reducing its effectiveness in a single-scenario evaluation. In Index-4, the student VLM model is first fine-tuned on simulated data and then fine-tuned on real-world data. This sequential fine-tuning strategy yields slight improvements compared to Index-2. The final experiment corresponds to our proposed model, which achieves an ADE of 1.44 and a collision rate of 1.58, representing reductions of 10.0% and 8.7%, respectively, compared to Index-4.

Table 3: Ablation studies are conducted on key design elements of the proposed method using the nuScenes-VLM and simulation datasets. The results demonstrate the effectiveness of the proposed design. The baseline large language model (LLM) used for comparison is LLaVA-v1.5 (7b).

| Index | Module Component Description | ADE ↓ | Collision Rate (%) ↓ |
|---|---|---|---|
| 1 | teacher VLM model baseline: Simulation dataset trained, real-world dataset tested | 17.33 | – |
| 2 | student VLM model baseline: Real-world dataset trained, real-world dataset tested | 1.66 | 1.80 |
| 3 | student VLM model baseline: Simulation dataset and Real-world dataset mixed trained | 1.84 | 1.96 |
| 4 | student VLM model baseline: Simulation dataset finetune firstly, then finetune with Real-world dataset | 1.60 | 1.73 |
| 5 | Proposed (LLaVA-7b) | 1.44 | 1.58 |

### 4.4 Qualitative Evaluation

We present additional qualitative results to further substantiate the superior performance of our model. Qualitative results are shown in fig. 2 and fig. 3. fig. 2 depicts a scenario in which the vehicle continues to move while the traffic light is green. fig. 2 illustrates a scenario involving the avoidance of potential risks. Guided by our end-to-end prompt instruction, the method generates the expected CoC answers and corresponding planning trajectories.

## 5 Conclusion

This paper introduces an end-to-end autonomous driving method based on a vision-language model (VLM) that transfers long-tail and challenging cases handling capabilities from simulated data to real-world deployment. The method comprises two baseline models and one discriminator. The two baseline models separately incorporate text instructions and sensor data from simulated and real-world datasets, respectively, to output CoC answers and future trajectories. The discriminator employs adversarial learning to enhance the handling of uncommon scenarios and transfers this knowledge to the student VLM model. Moreover, the overall training process accelerates convergence and achieves promising results. Finally, the effectiveness of the proposed approach is validated on the nuScenes-VLM dataset.

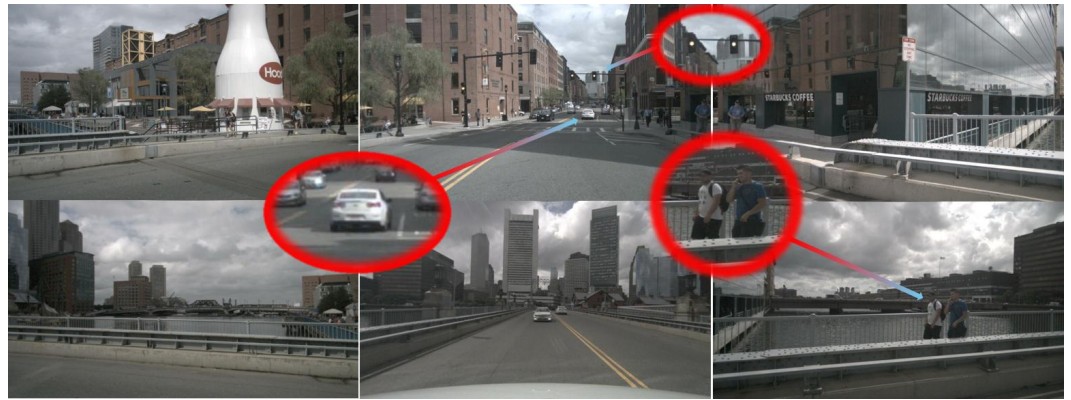

Temporal Instruction: In the previous frame, a white car ahead is moving away from the ego vehicle, and the vehicle continues in its current driving state.
End-to-end Prompt: What are the objects around the ego vehicle? What is the moving status of this object? What is the next action of the closest object? Will thread ego vehicle's safety? What are safe actions to take for the ego vehicle? Predict the future motion of the ego vehicle.

End-to-end Answer: There is a white car in front of ego vehicle, the coordinate is <CAM_FRONT, 1009, 486, 1074, 527>. The write car <CAM_FRONT, 1009, 486, 1074, 527> is accelerate moving away. There is a traffic light ..., There are two people .... In front of ego vehicle, there is no safety threat, the ego vehicle should keep going at the same speed. The future trajectory is (4.7, -0.7), (7.4, -1.3)....
Cache Temporal Information: There is no safety threat, the vehicle maintains its current speed.

Figure 2: Qualitative results. Ego vehicle passing through a traffic light.

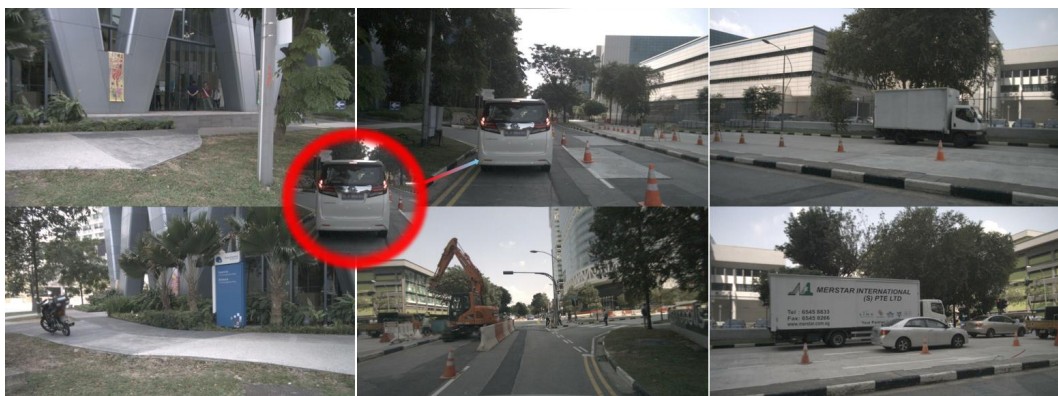

Temporal Instruction: From last frame, we find a white car is decreasing in front of ego vehicle, ego vehicle is slow down.
End-to-end Prompt: What are the objects around ego vehicle? what is the moving status of this object? What is the next action of the closest object? will thread ego vehicle's safety? What are safe actions to take for the ego vehicle? Predict the future motion of ego vehicle.

End-to-end Answer: There is a white car in front of ego vehicle, the coordinate is <CAM_FRONT, 551, 409, 909, 739>. The write car <CAM_FRONT, 551, 409, 909, 739> is braking. In front of ego vehicle, there is a threat in front of ego vehicle, ego vehicle should slow down and proceed with caution. The future trajectory is (0.3, -0.0), (0.9, -0.1)....
Cache Temporal Information: There is a threat in front of ego vehicle, ego vehicle should slow down and proceed with caution.

Figure 3: Qualitative results. A scenario in which a car brakes in front of the ego vehicle.

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

# A   Experiment Appendix

## A.1   More Ablation Study

### A.1.1   Effectiveness of Projection

We also train our module using different projection methods, as shown in Table 4. In this experiment, we replace the original projection module, which consists of several MLP layers, with the Q-former from BLIP-2 [43] to evaluate the effectiveness of our projection design. As shown in the table, the MLP-based projection slightly outperforms the Q-former-based projection. To further analyze this phenomenon, we identify two possible reasons: (1) the Q-former contains significantly more parameters than the MLP, making it better suited for large-scale datasets; and (2) the Q-former compresses image features into 256 tokens, which may result in the loss of fine-grained visual information, such as object positions, that is crucial for autonomous driving tasks.

Table 4: An ablation study is conducted to evaluate the effectiveness of the projection module on the nuScenes-VLM dataset. The study compares MLP-based projection with Q-former-based projection. Notably, the proposed model (LLaVA-7b) is implemented using an MLP-based projection module.

| Name | ADE ↓ | Collision Rate (%) ↓ |
|---|---|---|
| Proposed with Qformer (LLaVA-7b) | 1.47 | 1.60 |
| Proposed (LLaVA-7b) | 1.44 | 1.58 |

### A.1.2   Effectiveness of Temporal Information

As shown in Table 5, we conduct experiments on the nuScenes-VLM dataset to assess the effectiveness of incorporating temporal information. As described in the model section, our method filters previous answers, embeds them, and inputs them into the CoC VLM. In this experiment, we remove the temporal information aggregation process. As shown in the first row of Table 5, this modification results in a slight performance drop in both ADE and collision rate.

Table 5: Ablation study on the effectiveness of temporal information. The experiment uses the nuScenes-VLM dataset and the simulated dataset.

| Name | ADE ↓ | Collision Rate (%) ↓ |
|---|---|---|
| Proposed without Temporal Information (LLaVA-7b) | 1.53 | 1.64 |
| Proposed (LLaVA-7b) | 1.44 | 1.58 |

### A.1.3   Long-tail Performance

We conducted a more in-depth analysis to demonstrate our model's ability of handle uncommon cases. Specifically, we split the challenging subset of the nuScenes dataset (comprising 122 scenes) as our test set, while the remaining, easier cases were used for training. Here are challenging scene examples: Scene-0026_379: The ego vehicle is intercepted by a construction worker to give way to a construction truck approaching from the left. Scene-0046_568, Scene-0094_948, Scene-0131_1153, Scene-0162_1556: Pedestrian intrusions. Scene-0150_1358: A construction worker blocks the road using traffic cones or water barriers in front of the ego vehicle. Scene-0201_1978: The ego vehicle is obstructed by a car attempting to park. We trained and evaluated our model using this challenging subset. The performance results are presented below, as shown in Table 6.

Table 6: An ablation study of handling uncommon cases.

| Name | ADE ↓ | Collision Rate (%) ↓ |
|---|---|---|
| Our Student VLM model baseline | 2.08 | 2.16 |
| Straight Regressor Baseline | 2.89 | 3.61 |
| Proposed (LLaVA-7b) | 1.73 | 1.92 |

The results show that our proposed model achieved improvements of 0.35 and 0.24 compared to our student baseline in the new split dataset.

Furthermore, to analyze the performance on easier cases (e.g., straight-line driving), we designed a dummy regressor baseline that always predicts a straight trajectory. This baseline helps quantify how much of the evaluation performance on the new nuScenes split can be attributed to genuine methodological improvements. The dummy regressor, termed the Straight Regressor Baseline, is based on our method. To generate a straight trajectory, we ignore the y-axis and predict only the x-axis trajectory values (whereas existing methods predict both (x, y) coordinates). We trained this Straight Regressor Baseline, and the results are presented in Table 6. This demonstrates that the new split of the nuScenes dataset, which includes challenging scenarios, effectively validates our strategy's ability to transfer challenge-handling capabilities to real-world models.

### A.1.4 Closed-loop Evaluation

We employed NeuroNCAP [44] as the closed-loop simulator, as it supports the nuScenes dataset and provides pretrained rendering model checkpoints, making it well-suited for our method. We cloned the NeuroNCAP [44] and NeoRAD-Studio [45] repositories, replaced the example model (UniAD) with our own pretrained model from the main experiment, and integrated our checkpoints. Additionally, we downloaded the NeuroAD weights, modified the evaluation script accordingly, and conducted the closed-loop evaluation. Since NeuroNCAP offers a standardized benchmark and evaluation metrics, and is commonly used by other methods, we followed its recommended configuration. We evaluated our method on the suggested scenarios (e.g., scene-099, scene-0103, etc.). As shown in the table below, our proposed method outperforms UniAD, VAD, and our student baseline trained on the nuScenes dataset without adversarial transfer learning.

Table 7: An ablation study of closed-loop.

| Model | NeuroNCAP Score avg | Collision Rate (%) avg |
|---|---|---|
| UniAD | 1.84 | 68.70 |
| VAD | 2.75 | 50.70 |
| Our Student VLM model baseline | 3.07 | 48.83 |
| Proposed (LLaVA-7b) | 3.32 | 45.26 |

### A.1.5 Datasets

During pre-training, the teacher and student VLMs are trained separately using simulated and real-world datasets, respectively. The simulated dataset, referred to as CARLA-VLM, is collected using the CARLA Leaderboard v2 simulator and comprises 61.6% normal scenes and 38.4% challenging scenarios (e.g., traffic jams, near-miss vehicle interactions, and pedestrian intrusions). The real-world dataset used is prepared from the publicly available nuScenes dataset. To accelerate training, we initialize both models with checkpoints from LLaVA-v1.5 [40]. Each model is fine-tuned using image frames, temporal instructions, and end-to-end prompts, enabling the Chain-of-Causality (CoC) VLMs to align instruction, visual, and temporal information effectively. Efficient training is further facilitated by sampling frames at fixed intervals and applying random temporal shifts as augmentation. We use a real-world dataset (nuScenes-VLM) and a simulated dataset (CARLA-VLM) to pre-train the student and teacher VLM models, respectively.

nuScenes-VLM dataset: The student VLM is pre-trained on the nuScenes-VLM dataset, which is derived from the publicly available nuScenes dataset and enriched with textual prompts. We follow the official train/validation split provided by nuScenes.

CARLA-VLM dataset: We use this dataset as our simulation dataset to pre-train teacher VLM model. This dataset is configured with settings similar to nuScenes and annotated with textual prompts consistent with those in the nuScenes-VLM dataset. To enhance dataset transparency, we provide the statistical distribution of scenarios within the CARLA-VLM dataset. A new supplementary table presents detailed counts for each scenario category. We believe these additions improve the clarity of our dataset description and enable a more rigorous evaluation of the model's performance, particularly in challenging driving scenarios.

We have provided the statistical distribution of the simulated challenging scenarios to clearly characterize the dataset. A new supplementary table presents detailed counts for each scenario category. We

hope these additions enhance the transparency of our dataset and enable a more rigorous assessment of the model's performance in handling rare but critical driving events, as shown in Table 8.

Table 8: CARLA Scenarios Statistical Distribution.

| Scenarios Category | Proportion(%) | Clips Count(total 600) |
|---|---|---|
| Normal (Straight, Left, Right Turn) | 61.6 | 370 |
| Pedestrian Intrusion | 4.7 | 28 |
| Fog | 6.7 | 40 |
| Rain | 7.8 | 47 |
| NearMiss Vehicle Interactions | 2.7 | 16 |
| Traffic Jam | 4.5 | 27 |
| Traffic Accident | 1.8 | 11 |
| Vehicle Cut In | 2.2 | 13 |
| Opposite Vehicle Intrusion | 0.8 | 5 |
| Vehicle U-Turning | 0.5 | 3 |
| Construction Obstacle | 4.0 | 24 |
| Bicycle Intrusion | 0.5 | 3 |
| Lane Merge | 1.2 | 7 |
| No Traffic Light Intersection | 0.7 | 4 |
| Turn Left and Merge In | 0.3 | 2 |

### A.1.6 N frames Fusion

We have conducted an experiment using two cached frames, with the results presented in Table 9. The findings indicate that, compared to using one cached frame, there is no significant improvement in ADE or Collision Rate. Therefore, we have chose to use a single cached frame to reduce the number of input tokens.

Table 9: Ablation study on N frame temporal information.

| Model | ADE ↓ | Collision Rate (%) ↓ |
|---|---|---|
| With 1 Frame | 1.44 | 1.58 |
| With 2 Frames | 1.47 | 1.56 |

### A.2 Implementation Details

We conduct our experiments using nuScenes dataset and a simulator dataset collected from CARLA. Our approach adopts the CoC VLM architecture as the baseline for both the teacher and student VLM models, which are pre-trained on the simulation and real-world datasets, respectively. The LLM Brain component is initialized using the LLaVA pre-trained model [41]. Given the complexity of the training process and the associated convergence challenges, we adopt a multi-step training strategy. To prevent convergence to suboptimal local minima, we utilize a large batch size during training. Furthermore, different optimization algorithms are applied to the various modules to improve training performance. All experiments are conducted on eight NVIDIA H100 GPUs. To reduce computational resource requirements, we fine-tune our model using the LoRA method [46].

Before adversarial training, we load the pre-trained models for both the teacher and student VLM models. The adversarial training process consists of multiple steps, incorporating our unique back-propagation strategies.

Step 1:

Forward-Propagation: We freeze both the teacher and student VLM models and train only the discriminator. Simulation and real-world data are fed into the two VLMs, respectively, and the

resulting features are passed to the discriminator to outputs the logits. Backward-Propagation: Using the discriminator's output logits, we compute the loss and perform back-propagation. Since only the discriminator is being trained at this stage, we update only its parameters while keeping both VLM models frozen.

Step 2:

Forward-Propagation: In this process, only the student VLM and the discriminator are involved. Both real-world and simulation data are input into the student VLM, and the resulting features are passed to the discriminator to produce logits.

Backward-Propagation: This step introduces our novel contribution. We first back-propagate the discriminator loss through the discriminator without updating its parameters. Then, the propagated gradients, along with the VLM-specific losses, are used to update the student VLM model. This strategy reduces convergence instability and accelerates the adversarial training process.

### A.3 Adversarial Training Loss

Unlike existing methods, our approach incorporates additional distinct loss functions. During Step 1 of adversarial training, the discriminator loss is incorporated. This enables the discriminator to distinguish between the two data distributions, defined as follows:

$$l_{step1} = l_d \tag{12}$$

In Step 2, both the discriminator loss and the VLM autonomous driving loss are utilized, as shown below:

$$l_{step2} = l_{VLM} + l_d \tag{13}$$

However, as described in the Model Training section, our discriminator only passes the gradient parameters, and does not update their model parameters.

### A.4 Evaluation Metrics

To evaluate the proposed method, we employ two categories of metrics: language evaluation metrics and planning evaluation metrics.

#### A.4.1 Language Evaluation Metrics

In our experiments, several standard metrics are used to assess language performance.

**SPICE.** This is a prevailing metric used in VQA and image captioning, to evaluate the structure similarity of predicted texts with ground truth while ignoring the semantic meanings [47]. In detail, it parses the text into a syntactic dependency tree using probabilistic context-free grammar, then maps the dependency tree into a scene graph in a rule-based manner. The scene graph describes the objects, attributes, and their relationship in the original text, and the SPICE score is computed as the F-score of the generated scene graphs from prediction and ground truth [47].

**GPT Score.** We employ GPT Score [48] to measure the semantic alignment of answers and complement the SPICE metric. Specifically, the question, the ground truth answer, the predicted answer, and a prompt asking for a numerical score of the answer. GPT Score is a metric provided by ChatGPT. Traditional metrics mainly assess word-level performance and may not capture semantic nuances, potentially yielding unexpected evaluation outcomes. Leveraging ChatGPT's robust reasoning capabilities, we employ it to gauge prediction quality and derive a more rational score [48].

**BLEU.** Bilingual Evaluation Understudy (BLEU) is used to measure the n-grams between prediction and ground truth, and is sensitive to the word order. The n ranges from 1 to 4 in our experiment. With higher precision indicating a better match, The BLEU score is between 0 and 1, where 1 represents a perfect match and 0 represents the opposite [49].

**ROUGE_L.** Recall-Oriented Understudy for Gisting Evaluation-Longest Common Subsequence (ROUGE_L) calculates the precision and recall with the longest common sub-sequence, which utilizes the n-grams policy similar to BLEU, but mainly based on recall [50].

**CIDEr.** Consensus-based Image Description Evaluation (CIDEr) encodes the frequency of n-grams appearing in the text, calculates the weight of each n-gram through TF-IDF, represents the sentence in vector form using n-grams, and then calculates the cosine distance of the TF-IDF vector between the two text to measure their similarity [51].

Furthermore, we also use Accuracy and Match as suggested by [42] to evaluate our method.

#### A.4.2 Planning Evaluation Metrics

**ADE.** Average Displacement Error (ADE) is used to measure the performance of the planning results, it indicates the average L2 distance between the labeled ground truth trajectories and predicted trajectories.

**Collision Rate.** This metric is used to compute the ratio of evaluation frames that collides with objects in over all evaluation frames.

Notably, these metrics follow the VAD [2] settings, it will consider the error/collision rate as an average over 0.5, 1, 1.5, 2, 2.5, 3 seconds, in another word, it uses average over average strategy.

## B   Data Generation

### B.1   Data Collection

In our approach, the nuScenes dataset [52] is used as the real-world dataset. This dataset is collected using multi-view cameras and LiDAR, with annotations provided for each key frame. A simulation dataset is also collected using the CARLA Leaderboard v2 simulator [10]. The same configuration as nuScenes is adopted, including multi-view cameras, labels, and HD maps. The data are segmented into clips, consistent with the nuScenes format. In contrast, numerous challenging scenarios are simulated, such as accidents, traffic violations, and pedestrian trespassing. These scenarios are designed to enhance the generalization capabilities of the student VLM model.

### B.2   CoC Answer Generation

VLM-based autonomous driving methods require textual prompts paired with corresponding answers. Our method takes data clips as input, following the approach suggested by [42], with several enhancements. We augment the data with specific object descriptions around the ego vehicle, including orientation (azimuth), pixel bounding box coordinates (2D coordinates of the object in the camera view), and dimensions (height and width of the bounding box, which serve as distance cues). We refine the end-to-end question/answer templates using two main strategies: employing classification-based multiple-choice questions for improved stability, and structuring logical templates to coherently link tasks. These data are structured using chain-of-causality reasoning to extract end-to-end driving logic. Notably, both the real-world and simulation datasets are generated using the same prompt strategy and follow a consistent CoC answer format. These datasets are referred to as nuScenes-VLM and CARLA-VLM, respectively, in our paper.

## C   Motivations

Recently, LLM-based autonomous driving methods have employed two distinct types of datasets. One type consists of real-world datasets, such as nuScenes [52], while the other includes simulated datasets, such as those generated using the CARLA simulator [10]. Methods trained on real-world datasets suggest that models should be applied in real driving scenarios; hence, models trained on real data are considered optimal. However, these methods face several challenges. First, collecting and labeling real-world data is expensive. Second, real-world data lacks rare and challenging cases, such as pedestrian trespassing, traffic violations, and accidents. Finally, real-world data often lacks complex interactions and decision-making scenarios.

To address these limitations, other researchers have proposed models trained on simulated data. Simulations can be freely deployed with arbitrarily dynamic or static environments and can reproduce scenarios that are difficult or dangerous to capture in the real world, such as reckless driving, traffic collisions, and extreme lighting conditions. Moreover, simulations facilitate the collection of large-scale labeled datasets. However, a significant domain gap exists between simulations and real-world driving, limiting the applicability of simulation-trained models to real vehicles. Therefore, transfers strong handling capabilities from simulation to real-world deployment has become increasingly important.

Furthermore, to incorporate temporal information from previous frames, existing methods typically input sequences of images into LLM models [9]. This strategy significantly increases token length and computational resource consumption. Therefore, it is essential to develop a simple and efficient strategy for temporal aggregation.

## D   Why CoC

These Chain-of-Thought approaches often compromise the integrity of reasoning, resulting in fragmented and incoherent decision-making chains. In contrast, our Chain-of-Causality (CoC) method outputs entire CoC answer (End2End answer) that preserves internal causal linkages across the entire process. Specifically, it follows the sequence: T cached information $\rightarrow$ perception $\rightarrow$ prediction $\rightarrow$ planning $\rightarrow$ T+1 cached information. For example, if the ego vehicle was turning left in the previous frame, the model should detect an oncoming vehicle in the opposite lane, predict its trajectory, and generate an appropriate plan accordingly.

# E  More Related Works

## E.1  Traditional Autonomous Driving

Traditional autonomous driving methods are typically composed of four subtasks: detection, online mapping, prediction, and planning. Researchers have extensively studied each of these components, contributing a wide range of solutions tailored to individual challenges in the driving pipeline.

**Detection.** CenterNet [53] introduces two specialized modules: Cascade Corner Pooling and Center Pooling, to improve object detection performance. Differently, PointPillars [54] employs LiDAR point clouds to predict 3D bounding boxes. DETR3D [55], a DETR-inspired approach, utilizes 3D queries to extract image features and directly predict bounding boxes without requiring non-maximum suppression. Additionally, LSS [56] pioneers the use of depth prediction to construct a Bird's-Eye View (BEV) feature map for detection tasks. Similarly, PETR [57] enhances feature initialization by incorporating 3D positional encodings into image features, which are then processed via detection queries using a transformer-based mechanism. BEVFormer [11] further advances BEV-based detection through spatio-temporal transformers, significantly improving detection accuracy for autonomous driving.

**Online Mapping.** For constructing detailed road topology, researchers focus on accurately identifying lane locations and geometries. LaneNet [58] decomposes the lane detection task into edge proposal and line localization to mitigate confusion with visually similar objects. FIERY [59] performs dense segmentation on the BEV feature map to predict lane features. Furthermore, HDMapNet [60] transitions from dense segmentation to sparse map representations. Different from HDMapNet, VectorMapNet [61] directly predicts polylines in the BEV space, removing the need for heuristic post-processing. MapTR [62] introduces query-based representations for constructing lane topology. Inspired by these methods, MapExpert [63] further refines this approach by distributing specialized experts to handle heterogeneous map elements with varying geometric characteristics.

**Prediction.** Conventional prediction models use historical trajectories to forecast future movements. Early methods, such as FaF and IntentNet [64, 65], use neural networks for motion prediction. CoverNet and related works [66, 67] highlight the significance of dynamic behavior modeling. In addition, MultiPath [68] combines visual features extracted from cameras with convolutional neural networks to predict motion in BEV space. Different from methods above, VectorNet [69] introduces sparse representations for trajectory prediction. Some researchers also adopt sparse representations to forecast vectorized trajectories using transformer-based architectures [70, 71]. In contrast, some models adopt dense representations to predict occupancy and motion flow [59, 72]. Other methods, such as VIP3D and PIP [73, 74], integrate interactions with dynamic agents and static map elements to boost predictive performance. PnPNet [75] adds a tracking strategy that derives trajectory estimates from detection results. Recently, unified frameworks such as UniAD, VAD, and SparseDrive [1, 2, 20] jointly perform perception and prediction within a single model.

**Planning.** Planning represents the final stage in autonomous driving, where systems generate executable trajectories. ALVINN [76] is among the earliest neural-network-based planning models. More recent approaches integrate perception outputs to refine trajectory accuracy [77, 78]. Others, such as LookOut and ST-P3 [79, 80], incorporate rule-based optimizations. Notably, reinforcement learning-based methods introduce teacher VLM models to guide planning [81, 82]. PlanT [83] utilizes standard transformers to extract object representations for planning, while VAD2 [13] models planning actions as probabilistic distributions, thereby enhancing local planning precision and achieving strong closed-loop performance on benchmarks.

# F  Limitations

As LLaVA-7b is chosen as the baseline, inference computational cost and speed present challenges for deployment in online systems, limiting potential applications. Furthermore, maintaining two baseline models incurs high training costs. The complexity of the training process also poses optimization challenges, particularly during the initial epochs. Since the objective of our method is to transfer simulation capabilities to real-world performance, evaluation is conducted on real-world datasets. Consequently, closed-loop evaluation metrics are not assessed. [45, 44]

# G  Social Impacts

The development of our Visual Language Model (VLM) methods holds significant potential to enhance transportation systems by improving road safety and mitigating traffic congestion. These methods exhibit robustness and strong generalization capabilities across diverse driving environments.

Moreover, their interpretability could facilitate the transition of autonomous driving systems from black-box to white-box models, thereby accelerating deployment. However, because autonomous driving is closely tied to human safety, the current limitations in the safety and trustworthiness of our methods raise concerns about their widespread adoption. There is still a long way to go.

