# OpenReview forum: "CoC-VLA: Delving into Adversarial Domain Transfer for Explainable Autonomous Driving via Chain-of-Causality Visual-Language-Action Model"
_NeurIPS.cc/2025/Conference — NeurIPS 2025 poster_

### Official Review · Reviewer_a2Yp · 2025-06-24

**Clarity:** 2
**Significance:** 2
**Originality:** 2
**Rating:** 4
**Confidence:** 2

**Summary:**

This paper introduces an end-to-end autonomous driving framework that uses a vision-language model (VLM) to bridge the domain gap between simulation and real-world scenarios. It employs a teacher VLM (trained on simulated “long-tail” driving events) and a student VLM (trained on real data), both using a shared Chain-of-Causality VLM backbone. A novel transformer-based discriminator is trained adversarially to align the student’s features with the teacher’s, effectively transferring rare-case handling skills from simulation to the real domain. The Chain-of-Causality architecture extends a pre-trained multimodal model (LLaVA-1.5) with a temporal text adapter that feeds the previous time-step’s summarized output into the language model, enabling chain-of-thought reasoning for explainable decision making. On the nuScenes-VLM benchmark, the proposed approach significantly outperforms prior LLM-based driving methods (e.g. DriveLM) on both language understanding and planning metrics, demonstrating effective sim-to-real transfer and improved interpretability.

**Questions:**

**1**, In line 187, what is the scheme for choosing LLM outputs to encode temporal consistency? \
**2**, How to understand the data domain denoted as $X^S$ and $X^R$? Are they sensor data and image data from simulation and real-world? Then what does $y$ represent here?\
**3**, How is the distribution probability calculated in line 251? Do we explicitly know distribution $p^s$ and $p^r$?\
**4**, How does the convex function $\delta$ in line 280 involves in the distance/f-divergence calculation in equation 7? Is $\delta$ related with function $f$?\

**Ethical Concerns:**

["NO or VERY MINOR ethics concerns only"]

**Final Justification:**

Authors have answered all questions raised and concerns are properly addressed with more intuitive explanations and experiments. The training method prosed by the authors are interesting and seemingly novel for the Autonomous Viechle community. As long as authors improved their writing particularly the method part in their final version, I recommend boarder-line accept.

**Limitations:**

See above. The reviewer think this paper is interesting and novel with comprehensive experiments setting, but authors need to improve on the writing clarity: some of the technical details are missing but crucial for understanding. The reviewer will raise the score if concerns are properly addressed.

**Paper Formatting Concerns:**

No formatting concerns

**Quality:**

3

**Strengths And Weaknesses:**

##Strengths:

**1**, The use of adversarial techniques for domain adaptation is well-established; however, combining this with VLMs to facilitate explainable decision-making in autonomous vehicles is a novel contribution.\
**2**, The proposed Chain-of-Causality VLM introduces a mechanism to model causal relationships in driving scenarios, which is a unique approach to improving interpretability in autonomous systems.\
**3**, Figure 1 is concise and straight-forward, which elaborates the structures and data flow.\
**4**, The experiments are solid and detailed, building upon a comprehensive set of metrics.


##Weakness:

**1**, Line 105, citation need to be fixed. \
**2**, From line 191 to 196, the example is straight forward, but still a bit vague to understand what kind of content are needed in temporal instruction.\
**3**, The clarity of section 3.4 needs to be improved.  A list of questions can be found in below Question part.\
**4**, In section 3.5.1, it would be better to state data used for pre-training of both student and teacher VLM.\
**5**, As a stand alone contribution stated in line 90-91, this back-propagation strategy have not been found in section 3 for elaboration.\

---

> ### Author Rebuttal · Authors · 2025-07-31
>
> We sincerely appreciate the time and effort you have dedicated to reviewing our manuscript. We have carefully addressed your concerns in detail. We hope you might find the response satisfactory.
>
> Weakness 1:
>
> Thank you for pointing out this issue. We have revised the manuscript and corrected the citation for FusionAD accordingly.
>
> Weakness 2:
>
> We apologize for the unclear expression. The temporal instruction is primarily composed of three parts:
> 1.A fixed prompt: “**In the previous frame,**”
> 2.The main threat detected in the last frame, such as “**a white car ahead is moving away from the ego vehicle,**” or “**a car ahead is slowing down.**” If no threat is detected, the default is “**there is no safety threat.**”
> 3.The action taken by the ego vehicle, such as “**the vehicle continues in its current driving state,**” “**the vehicle is turning left,**” “**the vehicle is turning right,**” or “**the vehicle is slowing down.**”
> This temporal instruction clearly conveys the ego vehicle’s previous action, which is crucial for determining the next appropriate action, as prior actions often influence current decision-making.
>
> Weakness 3:
>
> We have revised Section 3.4 of the manuscript to improve its readability. The specific concerns have been addressed in detail in the question part (please refer to  Q1, Q2, Q3, and Q4). We hope that our responses to Q1, Q2, Q3, and Q4 could address your concerns.
>
> Weakness 4:
>
> We appreciate the reviewer’s insightful suggestion. In response, we have added a detailed explanation of the datasets used for pre-training both the student and teacher VLMs. Section 3.5.1 of the manuscript has been revised accordingly. The updated content is as follows:
>
> During pre-training, the teacher and student VLMs are trained separately using simulated and real-world datasets, respectively. The simulated dataset, referred to as CARLA-VLM, is collected using the CARLA Leaderboard v2 simulator and comprises 61.6\% normal scenes and 38.4\% challenging scenarios (e.g., traffic jams, near-miss vehicle interactions, and pedestrian intrusions). The real-world dataset used is prepared from the publicly available nuScenes dataset. To accelerate training, we initialize both models with checkpoints from LLaVA-v1.5 [40]. Each model is fine-tuned using image frames, temporal instructions, and end-to-end prompts, enabling the Chain-of-Causality (CoC) VLMs to align instruction, visual, and temporal information effectively. Efficient training is further facilitated by sampling frames at fixed intervals and applying random temporal shifts as augmentation.
>
> We use a real-world dataset (nuScenes-VLM) and a simulated dataset (CARLA-VLM) to pre-train the student and teacher VLM models, respectively.
> nuScenes-VLM dataset:
> The student VLM is pre-trained on the nuScenes-VLM dataset, which is derived from the publicly available nuScenes dataset and enriched with textual prompts (as illustrated in Appendix B.2). We follow the official train/validation split provided by nuScenes.
>
> CARLA-VLM dataset:
> We use this dataset as our simulation dataset to pre-train teacher VLM model. This dataset is configured with settings similar to nuScenes and annotated with textual prompts consistent with those in the nuScenes-VLM dataset.
> To enhance dataset transparency, we provide the statistical distribution of scenarios within the CARLA-VLM dataset. A new supplementary table presents detailed counts for each scenario category. We believe these additions improve the clarity of our dataset description and enable a more rigorous evaluation of the model’s performance, particularly in challenging driving scenarios.
>
> | Scenarios Category | Proportion(\%) |Clips Count(total 600)|
> |--|--|--|
> |Normal (Straight, Left, Right Turn) |61.6|370|
> |Pedestrian Intrusion | 4.7| 28|
> |Fog| 6.7| 40|
> |Rain|7.8| 47|
> |Near-Miss Vehicle Interactions|2.7| 16|
> |Traffic Jam |4.5|27|
> |Traffic Accident|1.8|11|
> |Vehicle Cut In|2.2|13|
> |Opposite Vehicle Intrusion|0.8| 5|
> |Vehicle U-Turning |0.5|3|
> |Construction Obstacle |4.0| 24|
> |Bicycle Intrusion |0.5| 3|
> |Lane Merge| 1.2|7|
> |No Traffic Light Intersection|0.7|4|
> |Turn Left and Merge In| 0.3| 2|
> Table Supp. CARLA Scenarios Statistical Distribution
>
> Actually, In all our experiments, we only used two datasets: the nuScenes-VLM dataset (real-world dataset) and the CARLA-VLM dataset (simulation dataset).
>
> Weakness 5:
>
> Thank you for your suggestion. We have revised the manuscript to enhance transparency. In our anonymous submission, our original intent was to explain the contribution of the back-propagation strategy in Section 3.5.2, with its corresponding data flow illustrated in Figure 1. We have now revised and clarified this content in Section 3.5.2 as follows:
>
> 3.5.3 Adversarial Training with Multi-Step Back-Propagation
>
> As illustrated in Fig.1, the adversarial training process involves multiple steps that incorporate our unique back-propagation strategy. Initially, the pre-trained teacher and student VLM models are loaded. The training procedure then proceeds in two steps:
>
> Step 1:
>
> Forward-Propagation
>
> In this forward propagation, we freeze the teacher VLM model and student VLM model, only train our discriminator. Simulation data and real-world data are fed separately into the teacher and student VLM models. The features extracted from both CoC VLMs are then passed into the discriminator.
> Back-Propagation
>
> After the discriminator outputs the final logits, we compute the discriminator loss and perform back-propagation. Since only the discriminator is trainable in this step, the gradients are back-propagated solely through the discriminator, and only its parameters are updated. The parameters of the teacher and student VLM models remain unchanged.
>
> Step 2:
>
> Forward-Propagation
>
> In this process, only the student VLM model and the discriminator are involved in the training pipeline. Both simulation and real-world data are fed into the student VLM model, and the resulting features are passed through the discriminator to produce logits.
> Back-Propagation
>
> This back-propagate step introduces our key contribution in terms of back-propagation. The discriminator loss is back-propagated through the discriminator, but without updating its parameters. Then, the gradients from this discriminator, together with the VLM-specific losses, are used to update only the parameters of the student VLM model.
>
> Q1:
>
> In line 187, our intention was to convey that temporal information from the previous frame is aggregated into the current frame to enhance temporal features. The term “temporal consistency” was used to indicate the continuity of information across the temporal dimension. To achieve this, we have employed a rule-based filter, this filter removes the reasoning part of End-to-end Answer, extract the final ego vehicle's action instruction such as "There is no safety threat, the vehicle maintains its current speed.", as described in lines 230. This filtering step reduces the input token length while preserving the most critical information from the previous frame.
>
> Q2:
>
> Dear Reviewer, we have carefully addressed your concerns as follows.
>
> $X^{S}$ and $X^{R}$ are macro-level representations denoting the data distributions of the simulated and real-world autonomous driving domains, respectively. A domain consists of an infinite set of data. Human only sample a finite subset from these distributions to form the training and testing datasets. In our paper, the datasets represent small sample sets drawn from both the virtual and real-world domains. Additionally, we use  $y$ to represent the labels/ground truth..
>
>
> Q3:
>
> Here is our explanation, our labels/GT corresponding to samples data $x^{s}$ and $x^{r}$ are mainly generated by GPT-4 and subsequently verified manually. These GTs, such as "There are two pedestrians ... In front of the ego vehicle..." are distribution samples from GPT4 outputs. Since natural language allows for diverse expressions of the same underlying information, it can have plenty of expression to describe the GT, for example "A pedestrian is on the left ...., another pedestrian is 10m away from ego...".
> To formalize this process, we use the statistical learning theory (also called VC-theory) to express general mathematical framework for estimating dependencies from samples. Different from most empirical methodologies (like deep learning) that often rely on intuitive, asymptotic, and/or biological arguments.
> We use an expression that returns a label $y$ for every input $x$, according to a conditional distribution $p(y|x)$, this is determined by GPT4 and manual verification. The expectation is:
> $$\bar y = \int dp(y|x))$$
> In our statistical field, we normally select enough samples to mitigate the bias. But in deep learning, like the existing LLM/VLM methods, they use few samples as the final expression. Thus, we explicitly highlight the distribution expression of language content generation.
>
> The distribution $p^s$ and $p^r$ are determined by GPT4 and manual verification.
>
>
>
>
> Q4:
>
>
>
> In our paper, the f-divergence is employed to measure the discrepancy between two domains. To enable optimization, the f-divergence function must be convex. Therefore, we adopt a convex function $\delta$ as our f-divergence.

---

> > ### Comment · Reviewer_a2Yp · 2025-08-04
> >
> > Thanks for author's effort in answering my questions, and some of my concerns are properly addressed. However, I'm still unclear about Question 4 since the answer provided doesn't seems to address my question with more detailed intuition.

---

> > > ### Author Response · Authors · 2025-08-05
> > >
> > > Dear reviewer,
> > >
> > > We sincerely appreciate the opportunity to discuss our manuscript and thank you for your thoughtful feedback. We apologize for the unclear explanation in our previous response to Question 4 and acknowledge that we  have misunderstood the intent of the question. Below, we provide a more detailed and clarified explanation.
> > >
> > > To address Question 4, we refer to Equation (7), which we divide into two parts: the left-hand side, representing a generalized formulation, and the right-hand side, representing a specialized approximation.
> > >
> > >
> > > **Left-hand Side (Generalized Expression)**
> > >
> > >
> > > In the equation (7), $D_{h}^{\delta}(X^{S}||X^{R}) $
> > > represents the discrepancy between two domains. This expression is generalized. The convex function $\delta$ also plays a generalized role, it is defined as a convex combination in parameter space, rather than over the sampled variables $x$ or $y$. Specifically, convex $\delta$ comprises a combination of two likelihood functions: one from the real-world domain and another from the simulation domain. Since both likelihood functions are convex, their convex combination remains convex. On the left-hand side of equation (7), we use an f-divergence or distance measure to quantify the domain gap. However, this expression is often intractable and must be explicitly approximated.
> > >
> > >
> > > **Right-hand Side (Specialized Expression)**
> > >
> > > The purpose of the right-hand side of equation (7) is to approximate the left-hand side using a finite number of samples. We give convergence bounds for algorithms that minimize a convex combination of two domains. Here, we use a large number of samples to approximate the computation of $D_{h}^{\delta}(X^{S}||X^{R})$. Specifically, we approximate the convex function $\delta$ through a convex combination of several explicit convex functions, such as, domain-specific models $ h_s \cdot g_s$, $ h_r \cdot g_r$. The convex combination can be convert to a convex function. For instance, a simple difference
> > >  $ h_s \cdot g_s - h_r \cdot g_r$ can act as an effective approximation of the divergence (sometimes, a simple subtraction could be used to replace complex divergence function). Therefore, we could use distance/ f-divergence to optimize this convex combination, thereby approximating the left-hand side definition.
> > >
> > >
> > > To sum up, the convex function $\delta$ is a generalized convex combination, it is convex function in the parameter space, rather than over the sampled variables $x$ or $y$. Thus, we did not design a specific $\delta$ function, instead, we use a convex combination composed of several explicit convex functions (such as, explicit models  $ h_s \cdot g_s$, $ h_r \cdot g_r$) to approximate $\delta$. These models, along with finite samples, enable us to optimize the domain discrepancy effectively  with distance/f-divergence.
> > >
> > >
> > > We are very much looking forward to hearing from you about any further feedback. We will be very happy to clarify and further concerns (if any).
> > >
> > >
> > > Wish you all the best.
> > >
> > > Warm regards
> > >
> > > Authors

---

> > > > ### Comment · Reviewer_a2Yp · 2025-08-05
> > > >
> > > > Thanks for the author’s answers. I don’t have any further questions and I will raise my score by 1

---

> > > > > ### Author Response · Authors · 2025-08-06
> > > > >
> > > > > Dear Reviewer, We truly appreciate your time and effort in reviewing our manuscripts and offering your valuable feedback. We thank you for allowing us to revise, which greatly moved us. We will carefully revise our manuscript.
> > > > >
> > > > > Once again, thank you for raising score.
> > > > >
> > > > > Wishing you great success with your paper!
> > > > >
> > > > > Warm regards
> > > > >
> > > > > Authors

---

### Official Review · Reviewer_orn8 · 2025-06-29

**Clarity:** 3
**Significance:** 3
**Originality:** 3
**Rating:** 4
**Confidence:** 3

**Summary:**

This paper proposes a Chain-of-Causality (CoC)-based transfer learning framework for autonomous driving, addressing the limitations of traditional methods that rely solely on either real-world or simulated data. By independently training two models and leveraging a teacher model (trained in complex simulated environments) to guide a real-world student model, the framework effectively utilizes simulated data. Additionally, the paper introduces Chain-of-Causality, which aggregates temporal information end-to-end via a text interface. Experimental results demonstrate that this method successfully achieves Sim2Real transfer in autonomous driving.

**Questions:**

The term "End2End answer" is quite ambiguous. While it seems to encompass textual descriptions of perception, prediction, and planning, is "end-to-end" the appropriate term? Would "CoC answer" be more accurate?

Have experiments been conducted using the previous N frames' "End2End answer" as cached temporal information?

The paper does not clearly differentiate between CoT (Chain-of-Thought) and CoC (Chain-of-Causality). In Omnidrive, this process is referred to as CoT, whereas the authors label it CoC. Clarifying the boundaries between these definitions would be helpful.

**Ethical Concerns:**

["NO or VERY MINOR ethics concerns only"]

**Final Justification:**

BA

**Limitations:**

Yes

**Quality:**

2

**Strengths And Weaknesses:**

Strengths:

The paper presents a novel perspective, and investigating the Sim2Real problem in autonomous driving holds significant research value.

The experiments are thorough, the analysis is well-reasoned, and the study offers strong analytical merit.

Weaknesses:

Figure 1 contains excessive lines, making the description overly complex and hindering readability.

The spacing between paragraphs appears inconsistent, and the excessive blank space before the "Contribution" section in the Introduction seems unnecessary.

The structure of Chapter 3 is somewhat unusual. Sections 3.2 and 3.3 could be condensed under "Methods" rather than being separate subsections, and "Model Training" might be better placed in Chapter 4 instead of Chapter 3.

The formatting of tables is inconsistent. For example, Table 1 lists each metric in separate columns, whereas Tables 2 and 3 do not.

---

> ### Author Rebuttal · Authors · 2025-07-31
>
> Thank you for the great efforts and valuable comments. We have carefully addressed the main concerns in detail. We hope you might find the response satisfactory.
>
> Weakness 1
>
> Response: Thank you for pointing this out. Specifically, we have revised Figure 1 by merging the flow lines to make the illustration more concise and clear. Additionally, we have moved the complex forward and backward propagation expression to the appendix (now shown as Figure 4 in our manuscript) and provided the corresponding explanations below:
> Before adversarial training, we load the pre-trained models for both the teacher and student VLM models. The adversarial training process consists of multiple steps, incorporating our unique back-propagation strategies.
>
> Step 1:
>
> Forward-Propagation:
> We freeze both the teacher and student VLM models and train only the discriminator. Simulation and real-world data are fed into the two VLMs, respectively, and the resulting features are passed to the discriminator to outputs the logits.
> Backward-Propagation:
> Using the discriminator's output logits, we compute the loss and perform back-propagation. Since only the discriminator is being trained at this stage, we update only its parameters while keeping both VLM models frozen.
>
> Step 2:
>
> Forward-Propagation:
> In this process, only the student VLM and the discriminator are involved. Both real-world and simulation data are input into the student VLM, and the resulting features are passed to the discriminator to produce logits.
>
> Backward-Propagation:
> This step introduces our novel contribution. We first back-propagate the discriminator loss through the discriminator without updating its parameters. Then, the propagated gradients, along with the VLM-specific losses, are used to update the student VLM model. This strategy reduces convergence instability and accelerates the adversarial training process.
>
> Weakness 2
>
> Response: We have removed the excessive blank space in the Introduction section, particularly before and after Figure 1.
>
> Weakness 3
>
> We have condensed the Sections 3.2 and 3.3 into the first paragraph of Sections 3. The modified content is shown below:
>
> As illustrated in Fig. 1, the proposed architecture consists of three components: a Teacher Visual Language Model (Teacher VLM), a Student Visual Language Model (Student VLM), and a Visual Language Model Discriminator. Both the Teacher and Student VLMs share a common base architecture, referred to as the Chain-of-Causality Visual Language Model (CoC VLM), which processes multi-view image pairs, end-to-end prompts, and historical instructions from previous frames to generate end-to-end outputs. The Teacher VLM is trained on simulated data to address diverse and rare scenarios, such as pedestrian trespassing, driving violations, and traffic accidents. In contrast, the Student VLM is trained on real-world data and serves as the final inference model, enabling the transfer of knowledge from simulated to real-world contexts.
>
> Additionally, we have relocated Subsection 3.5, Model Training, to the beginning of Section 4 as its first subsection. Following this reorganization, the overall structure is significantly clearer and more readable.
>
> Weakness 4
>
> We apologize for the oversight. We have revised the formatting inconsistencies in the tables by adding separate columns to Tables 2, 3, and 4, consistent with the format used in Table 1.
>
> Q1:
>
> Thank you for your suggestion. We agree that "CoC answer" is more appropriate than "End2End answer." Accordingly, we have revised the manuscript to replace all instances of “End2End answer” with “CoC answer.”
>
>
> Q2:
>
> We have conducted an experiment using two cached frames, with the results presented in Table 6. The findings indicate that, compared to using one cached frame, there is no significant improvement in ADE or Collision Rate. Therefore, we have chose to use a single cached frame to reduce the number of input tokens.
> Table 6. Ablation study on N frame temporal information.
> | Model               | ADE | Collision Rate        |
> | ------------------ | --------------- | ------------------ |
> | With 1 Frame | 1.44       | 1.58    |
> | With 2 Frame  | 1.47         | 1.56 |
>
> Q3：
>
>
> Existing Chain-of-Thought (CoT) autonomous driving methods typically rely on multiple rounds of dialogue, with each round performing a single-step reasoning process. For example, OmniDrive employs a sequence of prompts such as:
> 1. What traffic elements should I be aware of while driving in this area?
> 2. If I decide to accelerate and make a left turn, what could be the consequences?
> 3. What should be my next action given the current driving situation, and why?
>
> These Chain-of-Thought approaches often compromise the integrity of reasoning, resulting in fragmented and incoherent decision-making chains. In contrast, our Chain-of-Causality (CoC) method outputs entire CoC answer (End2End answer) that preserves internal causal linkages across the entire process. Specifically, it follows the sequence: T cached information → perception → prediction → planning → T+1 cached information. For example, if the ego vehicle was turning left in the previous frame, the model should detect an oncoming vehicle in the opposite lane, predict its trajectory, and generate an appropriate plan accordingly.

---

> > ### Comment · Reviewer_orn8 · 2025-08-04
> >
> > Thank you for providing a detailed response and the additional experimental results. After reviewing your rebuttal, I find that most of my concerns have been well addressed. The new results provide strong support for the claims made in the paper. Therefore, I will raise my score to borderline accept.

---

> > > ### Author Response · Authors · 2025-08-04
> > >
> > > Dear Reviewer,
> > >
> > > Thank you once again for your valuable comments and constructive suggestions. Your feedback has been extremely helpful in improving the quality of our manuscript. Your positive evaluation is very encouraging, and we will continue to revise and refine the manuscript accordingly.
> > >
> > > Wishing you great success with your paper!
> > >
> > > Wish you all the best.
> > >
> > > Warm regards
> > >
> > > Authors

---

### Official Review · Reviewer_hDQ3 · 2025-06-30

**Clarity:** 2
**Significance:** 3
**Originality:** 3
**Rating:** 4
**Confidence:** 5

**Summary:**

This paper proposes a novel end-to-end autonomous driving framework that aims to transfer long-tail handling capabilities learned in simulation to real-world applications via adversarial domain transfer. The authors introduce an architecture comprising a teacher VLM, a student VLM, and a discriminator. A central component of this architecture is the proposed "Chain-of-Causality Visual Language Model" (CoC VLM), designed to integrate temporal information and facilitate causal reasoning for decision-making.

**Questions:**

Regarding the CoC VLM architecture described on lines 179-182, could you please clarify the following:

1.  The 'filter' is described as a component to simplify LLM-generated responses and reduce token length. The paper later states that this filter is designed to "summarize and cache the end-to-end answer, converting it into a concise instruction for the subsequent frame". What is the specific mechanism or model used for this filter? Is it a rule-based system that extracts keywords, or another smaller language model?

2.  The 'dedicated end-to-end answer generation module' is mentioned as a way to streamline output formatting. Does this refer to a separate trainable module, or does it describe a specific structured prompting and decoding process?

**Ethical Concerns:**

["NO or VERY MINOR ethics concerns only"]

**Final Justification:**

Please see my comment below.

**Limitations:**

Please see the weakness section.

**Quality:**

2

**Strengths And Weaknesses:**

[Strengths]

The core idea of the paper is interesting and highly relevant. Leveraging simulation data to supplement the scarcity of rare and challenging cases in real-world datasets is a promising direction for developing more robust autonomous driving systems. The proposed adversarial transfer framework to bridge the sim-to-real gap for complex behaviors is reasonable.

[Weaknesses]

My major concern involves the experimental part of the manuscript. The evaluation methodology does not sufficiently support the central claims of the paper.

1.  The evaluation relies on macro-level metrics that may not validate the claimed improvement in handling long-tail scenarios. The paper reports the Average Displacement Error (ADE) and Collision Rate over the entire nuScenes-VLM test set. These overall metrics are likely dominated by a large number of simple, common scenarios (e.g., straight-line driving), which can dilute the performance results and mask the model's actual capabilities in critical, long-tail situations. To substantiate the claim of improved handling of uncommon cases, the authors must provide a more fine-grained analysis. For instance, they should isolate a subset of test clips containing specific long-tail events (e.g., 'pedestrian trespassing' or 'traffic violations' as mentioned in the simulation design ) and report performance metrics (ADE, Collision Rate) specifically on this challenging subset. Without this, the claimed improvements are not convincingly demonstrated.

2.  A significant limitation is the lack of closed-loop evaluation. The paper's evaluation is conducted in an open-loop setting, which assesses the model's ability to predict a trajectory but does not account for how the ego-vehicle's actions influence the environment's subsequent states. Closed-loop evaluation, where the model's output is fed back into the simulation to create a continuous interaction loop, is a much stronger and more realistic test of an autonomous driving system's performance and safety. The authors acknowledge this in the appendix, stating that 'closed-loop evaluation metrics are not assessed', but this remains a major weakness that impacts the perceived real-world viability of the proposed method.

3.  The paper lacks transparency regarding the statistical distribution of the simulated long-tail scenarios. While the supplementary material mentions that the simulation data is collected from CARLA Leaderboard v2's 39 challenging scenarios and aims for a 'balanced representation' to address the long-tail challenge, neither the main paper nor the supplement provides quantitative statistics on the final distribution of these cases. It is unclear what specific cases are included and at what frequency. This lack of transparency makes it difficult to assess the diversity of the training data and to fairly interpret the performance improvements. The effectiveness of the domain transfer heavily relies on the quality and distribution of these simulated events.

---

> ### Author Rebuttal · Authors · 2025-07-31
>
> We sincerely appreciate the time and effort you have dedicated to reviewing our manuscript. Your insightful comments have significantly improved the quality of our research. Below, we address your concerns point by point.
>
> Weaknesses 1:
>
>
> Response: Thank you for your thoughtful suggestion. We conducted a more in-depth analysis to demonstrate our model’s ability of handle uncommon cases. Specifically, we split the challenging subset of the nuScenes dataset (comprising 122 scenes) as our test set, while the remaining, easier cases were used for training. Here are challenging scene examples:
> Scene-0026_379: The ego vehicle is intercepted by a construction worker to give way to a construction truck approaching from the left.
> Scene-0046_568, Scene-0094_948, Scene-0131_1153, Scene-0162_1556: Pedestrian intrusions.
> Scene-0150_1358: A construction worker blocks the road using traffic cones or water barriers in front of the ego vehicle.
> Scene-0201_1978: The ego vehicle is obstructed by a car attempting to park.
> We trained and evaluated our model using this challenging subset. The performance results are presented below.
> | Model               | ADE | Collision Rate        |
> | ----------- | -------- | ----- |
> | Our Student VLM model baseline | 2.08       | 2.16           |
> | Proposed (LLaVA-7b)  | 1.73         | 1.92 |
> The results show that our proposed model achieved improvements of 0.35 and 0.24 compared to our student baseline in the new split dataset.
>
> Furthermore, to analyze the performance on easier cases (e.g., straight-line driving), we designed a dummy regressor baseline that always predicts a straight trajectory. This baseline helps quantify how much of the evaluation performance on the new nuScenes split can be attributed to genuine methodological improvements.
> The dummy regressor, termed the Straight Regressor Baseline, is based on our method. To generate a straight trajectory, we ignore the y-axis and predict only the x-axis trajectory values (whereas existing methods predict both (x, y) coordinates). We trained this Straight Regressor Baseline, and the results are presented below.
> | Method            | ADE | Collision Rate        |
> | ------------------ | --------------- | --------------- |
> | Straight Regressor Baseline   | 2.89    | 3.61    |
> This demonstrates that the new split of the nuScenes dataset, which includes challenging scenarios, effectively validates our strategy's ability to transfer challenge-handling capabilities to real-world models.
>
>
>
>
> Weaknesses 2:
>
>
> Response: We sincerely thank you for your suggestion. We employed NeuroNCAP [1] as the closed-loop simulator, as it supports the nuScenes dataset and provides pretrained rendering model checkpoints, making it well-suited for our method. We cloned the NeuroNCAP [1] and NeoRAD-Studio [2] repositories, replaced the example model (UniAD) with our own pretrained model from the main experiment, and integrated our checkpoints. Additionally, we downloaded the NeuroAD weights, modified the evaluation script accordingly, and conducted the closed-loop evaluation.
> Since NeuroNCAP offers a standardized benchmark and evaluation metrics, and is commonly used by other methods, we followed its recommended configuration. We evaluated our method on the suggested scenarios (e.g., scene-099, scene-0103, etc.). As shown in the table below, our proposed method outperforms UniAD, VAD, and our student baseline trained on the nuScenes dataset without adversarial transfer learning.
>
> | Model               | NeuroNCAP Score avg | Collision Rate avg  |
> | ------------------ | --------------- | ------------------ |
> | UniAD         | 1.84         | 68.70          |
> | VAD | 2.75       | 50.70           |
> | Our Student VLM model baseline | 3.07       | 48.83           |
> | Proposed (LLaVA-7b)  | 3.32         | 45.26 |
>
> Weaknesses 3:
>
> Response: We have provided the statistical distribution of the simulated challenging scenarios to clearly characterize the dataset. A new supplementary table presents detailed counts for each scenario category. We hope these additions enhance the transparency of our dataset and enable a more rigorous assessment of the model's performance in handling rare but critical driving events.
> | Scenarios Category | Proportion(\%) |Clips Count(total 600)|
> |--|--|--|
> |Normal (Straight, Left, Right Turn) |61.6|370|
> |Pedestrian Intrusion | 4.7| 28|
> |Fog| 6.7| 40|
> |Rain|7.8| 47|
> |Near-Miss Vehicle Interactions|2.7| 16|
> |Traffic Jam |4.5|27|
> |Traffic Accident|1.8|11|
> |Vehicle Cut In|2.2|13|
> |Opposite Vehicle Intrusion|0.8| 5|
> |Vehicle U-Turning |0.5|3|
> |Construction Obstacle |4.0| 24|
> |Bicycle Intrusion |0.5| 3|
> |Lane Merge| 1.2|7|
> |No Traffic Light Intersection|0.7|4|
> |Turn Left and Merge In| 0.3| 2|
> Table Supp. CARLA Scenarios Statistical Distribution
>
> Q1:
>
> We apologize for the unclear explanation. The filter referenced here is a rule-based system that removes the reasoning component of the End-to-End Answer and extracts only the final action instruction for the ego vehicle—for example, "There is no safety threat, the vehicle maintains its current speed." as described on line 230. This filtering process reduces the input token size while preserving the most critical information from the previous frame.
>
> Q2:
>
>
> The "dedicated end-to-end answer generation module" refers to our CoC VLM model. Unlike existing approaches that typically rely on multiple rounds of dialogue (they firstly generating a perception answer, then feeding it into the model to produce a prediction answer, and finally combining both to generate a planning answer), our model generates the end-to-end answer directly. This design simplifies the pipeline and streamlines output generation.
>
> [1] W. Ljungbergh, A. Tonderski, J. Johnander, H. Caesar, K. Åström, M. Felsberg, and C. Petersson, “Neuroncap: Photorealistic closed-loop safety testing for autonomous driving,” European Conference on Computer Vision (ECCV), 2024.
> [2] A. Tonderski, C. Lindström, G. Hess, W. Ljungbergh, L. Svensson, and C. Petersson, “NeuRAD: Neural rendering for autonomous driving,” in Proceedings of the IEEE/CVF Conference on Computer Vision and Pattern Recognition, 2024, pp. 14 895–14 904.598

---

> > ### Comment · Reviewer_hDQ3 · 2025-08-04
> > **Reply to rebuttal**
> >
> > I appreciate the efforts the authors made during the rebuttal. Most of my concerns are given further analysis. I will increase my score by 1.

---

> ### Author Response · Authors · 2025-08-04
>
> Dear Reviewer,
> We truly appreciate your time and effort in reviewing our manuscripts and offering your valuable feedback. We thank you for allowing us to revise, which greatly moved us. We will carefully revise our manuscript.
>
> Once again, thank you for raising score.
>
> Wish you all the best.
>
> Warm regards
>
> Authors

---

### Decision · Program_Chairs · 2025-09-17

**Decision:**

Accept (poster)

**Comment:**

The paper introduces a Chain-of-Causality (CoC) Visual Language Model and an adversarial transfer framework with teacher–student VLMs and a discriminator. The goal is to transfer rare-case driving behaviors from simulation to real-world data.

This paper proposes a novel integration of VLMs, adversarial transfer, and causal reasoning for autonomous driving and demonstrates strong sim-to-real transfer. Closed-loop evaluation (via NeuroNCAP) and additional analyses added during rebuttal strengthened claims and the revision addressed clarity, dataset transparency, and formatting issues.

The initial evaluation lacked fine-grained long-tail analysis and closed-loop tests (later added in rebuttal). Writing and structure in early versions were unclear; terminology (CoC vs. CoT, End2End answer) was confusing. Some theoretical explanations (e.g., convex f-divergence) were initially unclear but clarified after back-and-forth discussion.

Despite initial weaknesses in clarity and evaluation, the rebuttal substantially strengthened the paper.